# Carbon Fiber Reinforced PEEK Composites Based on 3D-Printing Technology for Orthopedic and Dental Applications

**DOI:** 10.3390/jcm8020240

**Published:** 2019-02-12

**Authors:** Xingting Han, Dong Yang, Chuncheng Yang, Sebastian Spintzyk, Lutz Scheideler, Ping Li, Dichen Li, Jürgen Geis-Gerstorfer, Frank Rupp

**Affiliations:** 1Section Medical Materials Science and Technology, University Hospital Tübingen, Osianderstr. 2–8, D-72076 Tübingen, Germany; xingting.han@student.uni-tuebingen.de (X.H.); Sebastian.Spintzyk@med.uni-tuebingen.de (S.S.); lutz.scheideler@med.uni-tuebingen.de (L.S.); ping.li@uni-tuebingen.de (P.L.); juergen.geis-gerstorfer@med.uni-tuebingen.de (J.G.-G.); Frank.Rupp@med.uni-tuebingen.de (F.R.); 2State Key Laboratory for Manufacturing System Engineering, School of Mechanical Engineering, Xi’an Jiaotong University, Xi’an 710054, China; yangdong2015@stu.xjtu.edu.cn (D.Y.); yang.chun.cheng@stu.xjtu.edu.cn (C.Y.)

**Keywords:** fused deposition modeling, polyether ether ketone, biocomposite, orthopedic implant, oral implant, mechanical properties, wettability, topography, biocompatibility, cell adhesion

## Abstract

Fused deposition modeling (FDM) is a rapidly growing three-dimensional (3D) printing technology and has great potential in medicine. Polyether-ether-ketone (PEEK) is a biocompatible high-performance polymer, which is suitable to be used as an orthopedic/dental implant material. However, the mechanical properties and biocompatibility of FDM-printed PEEK and its composites are still not clear. In this study, FDM-printed pure PEEK and carbon fiber reinforced PEEK (CFR-PEEK) composite were successfully fabricated by FDM and characterized by mechanical tests. Moreover, the sample surfaces were modified with polishing and sandblasting methods to analyze the influence of surface roughness and topography on general biocompatibility (cytotoxicity) and cell adhesion. The results indicated that the printed CFR-PEEK samples had significantly higher general mechanical strengths than the printed pure PEEK (even though there was no statistical difference in compressive strength). Both PEEK and CFR-PEEK materials showed good biocompatibility with and without surface modification. Cell densities on the “as-printed” PEEK and the CFR-PEEK sample surfaces were significantly higher than on the corresponding polished and sandblasted samples. Therefore, the FDM-printed CFR-PEEK composite with proper mechanical strengths has potential as a biomaterial for bone grafting and tissue engineering applications.

## 1. Introduction

Cranio-maxillofacial defects related to tumors, traumas, infections, or congenital deformities are highly challenging tasks for oral and maxillofacial surgeons to reconstruct [1,2]. When bone losses are too severe for human body routine mechanisms to regenerate, autologous grafts are the first considerations due to the simultaneous osteogenic, osteoinductive, and osteoconductive properties [3]. However, the shape of the donor sites, bone graft resorption, and infection restrict the application of autografts [4]. Currently, the most popular orthopedic/dental artificial materials are metals like titanium (Ti) and its alloys. These materials have many advantages, such as excellent biocompatibility, corrosion resistance, and mechanical strength [5]. However, there are some critical drawbacks of Ti, one of which is stress-shielding, which may occur at the interface between Ti and bone during load transfer and result in surrounding bone loss [6]. In addition, the radiopacity of Ti alloys in the CT and MR scan images and the release of harmful metal ions hinder the application of metals [7]. Due to the limitations observed in metallic biomaterials, polymers have been explored in recent years as potential alternative materials for bone replacement.

In the last few years, polyether ether ketone (PEEK) has been investigated widely in oral and cranio-maxillofacial surgery. Possible applications are dental implants, skull implants, osteosynthesis plates, and bone replacement material for nasal, maxillary, or mandibular reconstructions (Figure 1) [8,9,10,11]. PEEK is considered an alternative material for Ti due to its excellent biocompatibility, radiolucency, chemical resistance, low density (1.32 g/cm^3^), and mechanical properties resembling human bone. PEEK is a polyaromatic semi-crystalline thermoplastic polymer with an elastic modulus of 3–4 GPa (Table 1), which is much lower than that of Ti (102–110 GPa) and very close to the human trabecular bone (1 GPa) [8,12]. Moreover, the mechanical strengths of PEEK can be enhanced by the incorporation of other materials (e.g., carbon fibers) [8]. Normally, carbon fiber reinforced polyether ether ketone (CFR-PEEK) has an elastic modulus close to the human cortical bone (14 GPa), depending on the amount of reinforced carbon fiber and manufacturing methods. CFR-PEEK is considered as a promising candidate to replace metallic materials because of the inherited advantages of PEEK and improved mechanical properties [13,14].

Additive manufacturing (AM) is a layer-by-layer manufacturing method, fabricating specimens by fusing or depositing materials, such as metals, ceramics, plastics, or even living cells [16]. This technique is becoming popular in orthopedic surgery for fabricating patient-specific implants due to the low cost, the feasibility of complex architectures, and the short production time [17]. Selective laser sintering (SLS) has been the most popular AM technology for fabricating PEEK in the past decades [18,19]. Compared with SLS, fused deposition modeling (FDM) is one of the fastest growing three-dimensional (3D) printing methods due to the lower costs, easier use (filament vs. powder), and reduced risk of material contamination or degradation. Furthermore, it has increasingly been applied to the manufacturing of PEEK and its composites in recent years [20]. However, due to the semicrystalline structure and high melting temperature of PEEK (compared with other FDM filament materials like polylactic acid (PLA) and acrylonitrile butadiene styrene (ABS)), it is difficult to process PEEK objects by FDM printing and the process is liable to cause excessive thermal stress and thermal cracks [8,18]. Yang et al. and Wu et al. have already measured the mechanical properties of FDM-printed pure PEEK and found that compared with some traditional manufacturing methods (i.e., injection molding), FDM-printed PEEK had lower mechanical strengths, which were influenced by layer thickness, printing speed, ambient temperature, nozzle temperature, and heat treatment [21,22] FDM-printed PEEK composites, to the best of our knowledge, have not yet been studied.

Compared with Ti, the unmodified PEEK is bioinert and has limited osteoconductive properties, which may influence the osseointegration after implantation [23,24]. Surface topographical modification is one of the mechanical surface modification methods to increase the biological performance of cranio-maxillofacial implants [25]. Surface roughness may influence cell adhesion, and a roughened surface usually has a more extensive surface area which offers more binding sites for cell attachment [26]. Some studies have already analyzed the influence of surface roughness on the bioactivity of PEEK and its composites [26,27,28]. However, in these reports, PEEK and its composites were all manufactured by traditional techniques like milling, injection modeling, and compression molding. For the FDM-manufactured PEEK, most studies only analyzed the manufacturing process and mechanical properties of pure PEEK, without PEEK composites [17,18,22]. According to our knowledge, tests of the mechanical properties of FDM-printed CFR-PEEK are still lacking. Therefore, the aim of this study was to evaluate the mechanical properties and microstructures of PEEK and CFR-PEEK samples manufactured by FDM. Specific attention was paid to the question of whether the FDM printing process has introduced or produced toxic substances and to the influence of surface treatments on the cell adhesion on sample surfaces.

## 2. Materials and Methods

### 2.1. Sample Preparation and Surface Modification

#### 2.1.1. Sample Preparation

A 3D printing system for PEEK material and its composite from Jugao-AM Tech. Corp. (Xi’an, China) was used to prepare all test specimens. In the printing process, the PEEK material was heated and transformed into a semi-liquid state inside the nozzle. Then, the feedstock filament was forced to pass through the nozzle where it was melted and deposited in the form of a thin layer onto the platform. After one layer was finished, the platform went down along the *z*-axis equal to the pre-setting layer thickness. The desired geometry of the final complex objects was built layer-by-layer under the control of a computer. The extrusion temperature was set at 420 °C, and the printing speed was 40 mm/s. The bead width of each printing line was 0.4 mm, and the layer thickness was 0.2 mm (Table 2). Moreover, the PEEK filaments, as the material for 3D printing in this paper, were reprocessed from pellets (450G, VICTREX Corp., Thornton Cleveleys, UK), and 5% milled carbon fibers with a length of 80–150 µm and a diameter of 7 µm (Nanjing WeiDa Composite Material Co. Ltd., Nanjing, China) were chosen as the reinforcements. Before printing, a special fixative paper (Mingtai 3D Technology Co., Ltd., Shenzhen, China) was applied to the print bed for the objects’ adhesion and warping improvement. After printing the samples and cooling them down to room temperature, the samples were placed into a furnace (101-0 s, Shaoxing SuPo Instrument Corp., Shaoxing, China) for the heat treatment (tempering) process. After heating for 2 h at 300 °C, the samples were cooled down to room temperature to decrease shrinkage distortion and residual stress to obtain good mechanical performance of the parts.

The dimension of the PEEK and CFR-PEEK samples for testing the mechanical properties (tensile, bending, and compressive tests) was according to ISO standards. The dog-bone shape tensile testing specimens (90 mm × 5 mm × 4 mm) were printed according to ISO 527-1 standard [29], and the cuboid bending specimens (80 mm × 10 mm × 4 mm) were printed according to ISO 178 standard (Figure 2) [30]. According to ISO 604 standard, two sample groups were manufactured for testing compressive strength and compressive modulus, respectively [31]. The round-shaped PEEK and CFR-PEEK disc samples for the wettability, roughness, microstructure, and biological tests were produced with a diameter of 14 mm and a thickness of 1 mm.

Ti disc samples (Grade: 4, Straumann AG, Basel, Switzerland) with 15 mm diameter and 1 mm thickness were prepared from Ti sheet metal (Straumann AG, Basel, Switzerland) with a punching tool. The Ti samples were used as an additional control group for the wettability, roughness, and biological tests. All titanium discs underwent a surface treatment, which was consistent with the polished PEEK and CFR-PEEK.

#### 2.1.2. Sample Surface Modification

After printing, the PEEK and CFR-PEEK disc samples for wettability, roughness, microstructure, and biological tests were divided into three groups: as-printed (untreated) group, polished group, and sandblasted group (*n* = 12 per group). The untreated group included the directly printed samples, without any surface treatment. For the polished and sandblasted groups, all the discs were manually polished with a series of SiC abrasive papers up to P4000 (Buehler, Lake Bluff, IL, USA) by a polisher (Buehler, Coventry, UK). Then, the samples of the sandblasted group were further modified using a sandblasting machine (P-G 400, Harnisch + Rieth, Winterbach, Germany) with 120 µm alumina (Al_2_O_3_) particles (Cobra, Renfert, Hilzingen, Germany) under the pressure of 0.1 MPa at a distance of 50 mm for 15 s.

All Ti disc surfaces were modified using the same processes as for PEEK and CFR-PEEK samples by a series of SiC abrasive papers (1200, 2500, 4000 grit, Buehler, Lake Bluff, IL, USA) by a polisher (Buehler, Coventry, UK). After polishing, all the samples were cleaned with deionized (DI) water by an ultrasonic cleaner (Sonorex Super RK102H, Bandelin, Berlin, Germany) to remove residual Al_2_O_3_ particles on the sample surfaces.

### 2.2. Mechanical Properties Test

Mechanical tests were carried out using an electro-hydraulic servo mechanical testing machine (CMT4304, MTS Corp., Eden Prairie, MN, USA) according to ISO standards. ISO 527-1: 2012 (Plastics—Determination of tensile properties), ISO 178: 2010 (Plastics—Determination of flexural properties), and ISO 604: 2002 (Plastics—Determination of compressive properties) were applied for the tensile, bending, and compressive tests, respectively [29,30,31]. Six samples were tested for each batch with a 1 mm/min testing speed, and the test was performed at an ambient temperature of 20 °C.

### 2.3. Surface Characterization

To determine the surface morphology, samples of PEEK and CFR-PEEK from the untreated, polished, and sandblasted groups (*n* = 2 per group) were sputtered with a 20 nm thick Au–Pd coating (SCD 050, Baltec, Lübeck, Germany) and characterized by a scanning electron microscope (SEM) (LEO 1430, Zeiss, Oberkochen, Germany) at 200× and 2000× magnification.

The surface topography of the discs (*n* = 6 per group) was analyzed by a profilometer (Perthometer Concept S6P, Mahr, Göttingen, Germany). For each sample, 121 profiles were measured over a 3 mm × 3 mm area. The arithmetic mean height (Sa) and root mean square height (Sq) were calculated based on these topographies by software (MountainsMap Universal 7.3, Digital Surf, Besançon, France).

The water contact angle (WCA) was measured at room temperature on six samples per group using a drop shape analyzer (DSA 10-MK 2, Kruess, Hamburg, Germany). Drops of 2 μL of distilled water were deposited on the respective disc surfaces using an automatic pipette. After 20 s wetting time, the contact angle at the air–water–substrate interface was quantified from the drop geometry using DSA software (version 1.90.0.11, Kruess, Hamburg, Germany).

### 2.4. Biological Tests

Biological tests consisted of an extract test and a direct contact test to analyze the cytotoxicity and of the investigation of cell attachment to the different samples (*n* = 9 per group). Two materials (PEEK and CFR-PEEK) with three different surfaces each (untreated, polished, and sandblasted) were tested. In each test, *n* = 3 samples were used for each surface modification. All tests were performed three times in independent experiments. Directly before the biological tests, samples were ultrasonically cleaned with DI water for 15 min and sterilized with 70% ethanol and 100% ethanol (15 min each). Subsequently, the samples were dried on filter paper in a sterile workbench (Lamin Air HB2472, Burgdorf, Switzerland).

#### 2.4.1. Cell Culture

L929 fibroblasts (DSMZ GmbH, Braunschweig, Germany) were cultured in DMEM medium (21063-029, Gibco, Paisley, UK) containing 10% fetal bovine serum (FBS, Life Technologies, Carlsbad, CA, USA), 1% penicillin/streptomycin (15140-122, Life Technologies Co., Carlsbad, CA, USA), and 1% GlutaMAX (Life Technologies Co., Paisley, UK) in 75 cm^2^ sterile cell culture flasks (Costar, Corning, Tewksbury, MA, USA). The cells were maintained in an incubator under a humidified atmosphere with 5% CO_2_ at 37 °C. The DMEM culture medium was renewed twice a week. When cells reached confluence, Trypsin (GIBCO, Paisley, UK) was used to detach cells from the bottom of flasks, and 1/10 of the total cells were transferred into a new flask.

#### 2.4.2. Test for In Vitro Cytotoxicity

The in vitro cytotoxicity test of PEEK and CFR-PEEK was performed by an extract method based on ISO 10993-5 [32]. Extracts were derived from soaking the samples with DMEM cell culture medium for 24 h at 37 °C. The ratio between the sample surface area and extraction vehicle volume was 3 cm^2^/mL. In the meantime, the cells were precultured for 24 h. The seeding concentration of L929 cells was 30,000 cells/cm^2^ in 200 μL DMEM medium per well in a 96-well plate (Cellstar 655180, Greiner Bio-One, Frickenhausen, Germany). After 24 h, the culture medium was removed from the cells and replaced by 150 μL extracts obtained from the respective sample groups. Three concentrations of each extract were tested: (a) undiluted (150 μL extracts), (b) 1:3 diluted with medium (50 μL extracts + 100 μL medium), and (c) 1:10 diluted (15 μL extracts + 135 μL medium). Ti samples were used as the negative control, and copper (Cu) samples were used as the positive control. After culturing for an additional 24 h, the extracts in all groups were replaced by 100 μL fresh DMEM medium to avoid artifacts in the following assay caused by blue color in the Cu extracts. The cytotoxicity was quantitatively analyzed by CCK-8 assay (Dojindo Molecular Technologies, Inc., Rockville, MD, USA). The volume of CCK-8 solution added to each test well was 10 μL. After incubating for 2 h, the optical density (OD) value was measured by a microplate ELISA reader (Tecan F50, Tecan Austria, Groedig, Austria) at 450 nm wavelength. The metabolic activity of L929 cells in the different test groups in comparison to the negative control was calculated according to the following formula:Cell metabolic activity (%) = (OD_t_ − OD_b_)/(OD_nc_ − OD_b_) 100%,(1)
where the OD value is the absorbance value of the respective test group (OD_t_), blank control group (OD_b_), and negative control group (OD_nc_).

#### 2.4.3. Cell Adhesion and Spreading

L929 cells were seeded on PEEK, CFR-PEEK, and Ti samples in 12-well plates (REF 3512, Costar, Kennebunk, ME, USA) with a density of 30,000 cells/cm^2^ and incubated in 2.4 mL DMEM medium at 37 °C and 5% CO_2_. After incubation for 24 h, cell adhesion was terminated by rinsing with Hank’s balanced salt solution (HBSS, Biochrom AG, Berlin, Germany). Adhering cells were vital stained for 10 min in a solution of 25 μg/mL fluorescein diacetate (FDA) and 1.25 μg/mL ethidium bromide (EB) (Sigma-Aldrich Chemie GmbH, Taufkirchen, Germany) in HBSS. For each sample, a minimum of six typical surface areas of every magnification (25×, 100×, 200×, and 400×) was documented by an Optishot-2 fluorescence microscope (Nikon, Tokyo, Japan) equipped with a digital camera (550D, Canon, Tokyo, Japan). Cell adhesion and spreading were assessed by measuring the density of the vital-stained cells (cells/cm^2^) and the mean area of sample surface covered by cells (% of Ti) using a photo editing software (ImageJ, v1.8.0, National Institutes of Health, Bethesda, MD, USA).

### 2.5. Statistical Analysis

SPSS Version 21 (SPSS INC, Chicago, IL, USA) was used for analyzing the data. Shapiro–Wilk and Levene tests were applied to assess the assumptions of data normal distribution and homogeneity of variances. The results of the mechanical properties of each parameter were tested using the Student’s *t*-test of unpaired data with equal variance. One-way analysis of variance (ANOVA) was used for the cell density, and cell adhesion and spreading followed by Tukey post-hoc test (α = 0.05). The contact angle and roughness data were analyzed by Kruskal–Wallis analysis (α = 0.05) for the disobedience of the data normality or homogeneity of variances.

## 3. Results

### 3.1. Mechanical Properties

Table 3 shows the results of mechanical tests for the FDM-printed PEEK and CFR-PEEK samples. From Table 3, it is observed that the PEEK samples with reinforced carbon fiber had significantly better strengths than the bare PEEK in the tensile and bending tests (*p* < 0.05). As for the compressive test, there was no statistical difference between the two materials in compressive strength.

### 3.2. Surface Characterization

To understand how topological factors affect cell adhesion and spreading, the surface morphology of PEEK and CFR-PEEK composite was determined using SEM. Figure 3 presents the SEM images of untreated, polished, and sandblasted PEEK and CFR-PEEK samples. Printing borders, as shown in Figure 3a,d were formed on the surface due to the deposition between two printing lines. The clear peaks and valleys, which completely disappeared after polishing and sandblasting, could be identified on both untreated PEEK and CFR-PEEK sample surfaces. The polished surfaces displayed the smoothest morphology, although a few defects remained on the polished CFR-PEEK surfaces (Figure 3b,e). The surfaces of specimens, however, after sandblasting treatment possessed surface topography features in the micrometer scale with a homogeneous distribution of protuberances and cavities (Figure 3c,f).

Figure 4 illustrates the roughness of specimens of different groups. It is obvious that the untreated specimens displayed the roughest surfaces, both for PEEK and CFR-PEEK materials with the Sa value of 17.67 ± 5.7 µm and 32.36 ± 17.02 µm, which were significantly higher than the values of the polished and sandblasted groups (*p* < 0.05). The Sa values of sandblasted PEEK (0.85 ± 0.14 µm) and CFR-PEEK (0.97 ± 0.26 µm) samples were slightly higher than those of the polished surfaces (0.42 ± 0.26 µm and 0.67 ± 0.42 µm). In contrast, the polished Ti samples showed a very smooth surface (0.2 ± 0.04 µm), which was more homogenous compared with that of the polished PEEK and CFR-PEEK samples. The same trend could also be seen in the Sq data.

The result of contact angle analysis is shown in Figure 5. Data revealed that the untreated surfaces of pure PEEK reflected an obvious hydrophobic response to water with a mean contact angle of 105 ± 26°. The polished PEEK specimens exhibited a hydrophilic behavior (78 ± 3°). After sandblasting, the contact angle rose slightly (88 ± 7°), but the difference was not significant (*p* > 0.05). As for the CFR-PEEK samples, the untreated group also indicated the most hydrophobic sample surface (92 ± 12°) compared with polished (82 ± 5°) and sandblasted (75 ± 3°) specimens. Both PEEK and CFR-PEEK samples, whether with or without surface modifications, revealed a more hydrophobic response to water compared to Ti (51 ± 5°).

### 3.3. Cytotoxicity

Cell metabolic activity is expressed as a percentage of the mean OD value of cells cultured with extracts of the negative control (Ti), as displayed in Figure 6i,j. The data showed high cell viability in the cultures treated with 1:1, 1:3, and 1:10 extract concentrations of both tested materials, PEEK and CFR-PEEK, independent of the respective surface treatment (PEEK: untreated: 98 ± 23%, 106 ± 17%, 97 ± 22%; polished: 105 ± 25%, 106 ± 16%, 102 ± 15%; sandblasted: 106 ± 33%, 114 ± 27%, 98 ± 37%; CFR-PEEK: untreated: 98 ± 20%, 100 ± 28%, 97 ± 23%; polished: 102 ± 27%, 97 ± 31%, 105 ± 25%; sandblasted: 99 ± 38%, 100 ± 33%, 96 ± 23%). All extracts of PEEK and CFR-PEEK samples showed no toxicity after 24 h incubation. Cell viability in all cultures was significantly above the 70% level regarded as toxicity threshold according to ISO 10993-5 [32]. The results were confirmed by morphology analysis as seen in Figure 6a–h. Cells in the 100% extracts of the PEEK and CFR-PEEK groups exhibited a similar appearance as the negative control (Ti) group with distinct fibroblastic profiles. On the contrary, a large number of dead cells appeared in the positive group with a cell survival rate of 1 ± 2%.

### 3.4. Cell Adhesion and Spreading

Cell viability, attachment, and spreading were examined through a LIVE/DEAD staining assay, as shown in Figure 7. Compared with polished and sandblasted samples, untreated samples indicated more attached cells on the surfaces, both for PEEK and CFR-PEEK materials (Figure 7a–f). In addition, many cells attached in lines in the valleys resulting from the FDM manufacturing process (Figure 7a,d). Figure 7h,i reveals the quantitative cell density and quantification of the mean surface area covered by cells. Cell density on the sample surfaces of untreated PEEK and CFR-PEEK was significantly higher than on the corresponding polished and sandblasted groups (*p* < 0.05), where density was close to the Ti group. Moreover, the untreated groups showed higher cell coverage compared to the modified surfaces. The polished groups showed the lowest cell attachment for PEEK as well as CFR-PEEK samples.

Figure 8 shows the attached L929 cells around PEEK (Figure 8a–c), CFR-PEEK (Figure 8d–f), and Ti (Figure 8g) samples of the direct contact test after culturing for 24 h. The cells on PEEK and CFR-PEEK samples showed fibroblastic features and distinct profiles unaffected by the different materials and surface modifications. Moreover, the cell number was also similar to the negative control (Ti), which confirmed that the PEEK and CFR-PEEK materials were not toxic.

## 4. Discussion

This study aimed to investigate the mechanical properties of FDM-printed PEEK composite, the influence of manufacturing on the materials’ cytotoxicity, and the impact of surface topography and wettability on cell adhesion. To the best of our knowledge, there is currently no literature on these topics, whereas the manufacturing parameters and mechanical properties of FDM-processed bare PEEK and the SLS-printed PEEK composite have already been published elsewhere [18,21,33,34,35]. According to the manufacturing principles of FDM, only thermoplastic filaments can be used, like PLA, ABS, and PEEK [33]. However, it is a great challenge to fabricate ideal-performance PEEK objects through FDM equipment due to its high melting temperature (above 300 °C), high melting expansion, and especially the semicrystalline property, in particular for PEEK composites [22,34]. In this study, FDM-printed CFR-PEEK composite was successfully fabricated, and the mechanical properties were first measured. Moreover, the influence of the surface topography and roughness on biocompatibility and cell adhesion of FDM-printed PEEK and CFR-PEEK was also estimated for the first time.

The mechanical results indicated that the pure PEEK showed low strength in tensile, bending, and compressive tests. However, the addition of 5% carbon fiber into the PEEK matrix improved the mechanical strengths (Table 3), showing values similar to those of human cortical bone (elastic modulus: 14 GPa) [8] Normally, the mechanical properties of additively manufactured PEEK were obviously lower than the traditionally produced parts (i.e., injection molding) [21]. Although some studies have been done on PEEK composites by adding reinforcement fillers using SLS technology, the mechanical properties of FDM-printed PEEK composites were still insufficient, compared with their cast counterparts as a bone replacement material for severe cranio-maxillofacial defects [34,35]. The manufacturing conditions of the FDM process, such as layer thickness, printing speed, ambient temperature, nozzle temperature, and heat treatment, can produce a significant impact on the mechanical properties of PEEK samples [21,22]. In this study, the tensile strength of bare PEEK was 95.21 ± 1.86 MPa with an elastic modulus of 3.79 ± 0.27 GPa, which was comparable to the injection-molded pure PEEK (100 MPa and 4 GPa) [21]. While the tensile strength and elastic modulus of CFR-PEEK composites reached 101.41 ± 4.23 MPa and 7.37 ± 1.22 GPa, which were much higher than the injection-molded pure PEEK, the similar trend could also be seen in the bending and compressive tests. This result indicates that the printing conditions used in this study were suitable for PEEK and CFR-PEEK manufacturing. Deng et al. and Wu et al. have measured the mechanical properties of FDM-printed pure PEEK and found that the mechanical strength of printed PEEK samples was significantly lower than the traditionally produced objects, whereas in this study the values were quite similar [18,21]. One proper explanation for the excellent mechanical properties in this research is the application of post heat treatment (tempering). Theoretically, heat treatment methods can increase the degree of crystallinity and relieve the residual stress and shrinkage distortion, which will increase the mechanical performance of PEEK parts [22]. Therefore, the mechanical strengths of PEEK composite could be tailored by carbon fibers to mimic human cortical bone, thus avoiding stress shielding [8].

Polishing and sandblasting are common surface processing methods in dentistry to get a smooth or rough surface. However, the FDM-printed sample surfaces were much rougher compared with sandblasted ones, as shown in Figure 3 and Figure 4. This finding can be related to the working principle of FDM. Thermoplastic materials are extruded by the printing nozzle, which can move across the building platform in *x*- and *y*-axes, to generate a 2D layer line by line. Then, a 3D object is built up by melting the successive 2D layers together. The crosswise oriented, threadlike inner structure of the specimen results in some unfilled areas between lines and layers, and also in the original printing structures on sample surfaces [36]. In this study, the sandblasted samples showed slightly rougher surfaces than the polished ones. Compared with some previous studies, the sandblasting parameters (i.e., distance and pressure) in this research had to be set lower in order not to perforate the layer-by-layer manufacturing pattern [26,27,37]. In other studies, using traditional methods to fabricate PEEK and its composite samples like injection molding or milling, the interior of the blocks was homogenous without layers or unfilled areas. The samples in this study were produced using FDM technology, laying down objects in layers with a thickness of 0.2 mm. If a higher sandblasting pressure or closer distance were applied to modify the sample, the upper surface layer would be exfoliated (Figure 9). Therefore, based on the parameters used for sandblasting in this study, the sandblasted sample surfaces were slightly rougher than the polished ones, but not obviously different.

It is recognized that the surface wettability of biomaterials is important for their bioactivities, such as cell adhesion and spreading [38]. Therefore, the hydrophilicity of the samples was evaluated by the static sessile drop method, and the results are shown in Figure 5. Both PEEK and CFR-PEEK materials, before surface modification, represented a hydrophobic response to water (contact angle between 90–110°), which is typical for PEEK materials [8,39]. After polishing and sandblasting, both samples exhibited slightly hydrophilic behavior with contact angles below 90°. Commonly, wettability is closely related to the surface topography and chemical composition of a material [39]. The higher water contact angle in the untreated group in this study could be explained by the printing structures produced by FDM (Figure 3 and Figure 4). On highly roughened surfaces, the peaks and valleys prevent the water droplet from spreading on the surface, which can result in increased contact angles since the peaks and valleys on the sample surfaces constitute “geometrical barriers” for the droplet spreading [37,40]. According to the study undertaken by Ourahmoune et al., the surface morphology strongly influences the hydrophilic behavior of PEEK and its composites [37]. For the polished and sandblasted samples, since the differences of roughness values between these two groups were not obvious, the water contact angles were similar.

Due to its chemical inertness, PEEK provides inherent good biocompatibility, and this is also one of its advantages that favors its clinical use [8]. However, for the FDM-printed PEEK using a relatively new technology to fabricate PEEK using AM, studies focusing on the possible introduction of toxic substances during the printing process are still lacking, especially for its composites. According to ISO 10993-5, a reduction of cell viability by more than 30% indicates a cytotoxic effect [32]. In this study (Figure 6), the cell metabolic test of PEEK and CFR-PEEK samples showed that more than 96% of cells survived in all sample groups tested, independent of the respective surface modification. This result was comparable to the negative control group (Ti). The cytotoxicity results indicated that there were no toxic effects generated by the printing process. Moreover, after surface treatment, some carbon fibers were exposed on the surface of CFR-PEEK samples. However, this exposure has not led to increased cytotoxicity. Zhao et al. investigated FDM-printed pure PEEK and obtained a similar result that no toxic substances were introduced during the printing process [17].

Cell adhesion and spreading are closely related to surface properties, that is, composition, roughness, morphology, and wettability [41]. In addition to chemical composition, surface roughness and morphology play a critical role in the biological responses of biomaterial surfaces. In this study, the untreated PEEK and CFR-PEEK sample surfaces exhibited significantly more cell attachment than the polished and sandblasted samples, where the attachment level was close to the Ti surfaces. The as-printed PEEK and CFR-PEEK showed a higher cell density which might be due to the special 3D-printed structures. As shown in Figure 3a,d and Figure 4a,d, the clear ridges and valleys on the surfaces could be identified on both PEEK and CFR-PEEK sample surfaces. These special printing structures could enlarge the surface area significantly compared with polished and sandblasted surfaces. Significantly more spaces are available for cells to attach and spread on this geometrical morphology. For many engineering applications, a post-printing process is always needed to eliminate the manufactured structures [39]. However, to improve the cell attachment and spreading, a rough surface as generated by FDM seems beneficial, which could not be achieved by sandblasting. It was obvious that the cells accumulated in the surface grooves resulting from the manufacturing process (Figure 7a,d). Figure 4a,d showed the reconstructed 3D surface topographies of the as-printed PEEK and CFR-PEEK samples. The cells could slide into the valleys on the sample surfaces and attach there. As for both the polished and sandblasted surfaces, the originally printed surface structures were removed and the surfaces showed a lower cell density, but the cells appeared more homogeneously attached. After polishing and sandblasting, the exposure of carbon fibers on the surface of CFR-PEEK samples did not improve the cell attachment significantly. This finding confirmed that reinforced carbon fibers could improve the mechanical properties of FDM-printed PEEK, but would not influence the cytotoxicity and cell adhesion. In this study, the biological response of FDM-printed PEEK was investigated at a basic level, including cytotoxicity and cell adhesion. In future studies, more biological tests (e.g., in vitro cell metabolic activity, proliferation, and in vivo osseointegration) should be applied to evaluate bioactivities.

To sum up, the results indicate that the FDM-printed CFR-PEEK has excellent mechanical properties compared with the printed bare PEEK. In addition, no toxic substances were introduced during the FDM printing process. FDM technology can yield a highly roughened surface suitable for cells to attach. 

## 5. Conclusions

In this study, the mechanical properties of FDM-printed, carbon fiber reinforced PEEK composite were systematically studied for the first time, including tensile, bending, and compressive tests. The experimental results confirmed that samples printed from pure PEEK material showed mechanical properties comparable to traditionally manufactured PEEK objects, obtained by extrusion techniques for example. On the contrary, the printed CFR-PEEK specimen represented significantly improved mechanical properties compared to printed pure PEEK. FDM technology could be used to provide more satisfactory mechanical strength of PEEK and its composites. Therefore, it is an appropriate method for matching the mechanical properties of PEEK composites with carbon fibers to mimic human cortical bone and avoid stress shielding in clinical applications, like dental implants, skull implants, osteosynthesis plates, and bone replacement material for nasal, maxillary, or mandibular reconstructions. 

Additionally, the impact of the surface topography and roughness of FDM-printed PEEK and its composites on biocompatibility and cell adhesion was also estimated for the first time. Laboratory experiments here clearly showed that no toxic substances were introduced during the FDM manufacturing process of pure PEEK and CFR-PEEK. Surface treatments leading to partial exposure of the fiber compound in the bulk material did not lead to increased cytotoxicity. FDM-manufactured surfaces had highly rough topographies, which could not be achieved by typical dental sandblasting processes. This structure was more suitable for cells to attach and spread compared with polished and sandblasted surfaces, resulting in a cell density comparable to that on Ti sample surfaces. Although tests carried out in this study are limited, it is expected that the CFR-PEEK composite with its enhanced mechanical properties has great potential to be used as an orthopedic or dental implant material in bone repair, regeneration, and tissue engineering applications.

## Figures and Tables

**Figure 1 jcm-08-00240-f001:**
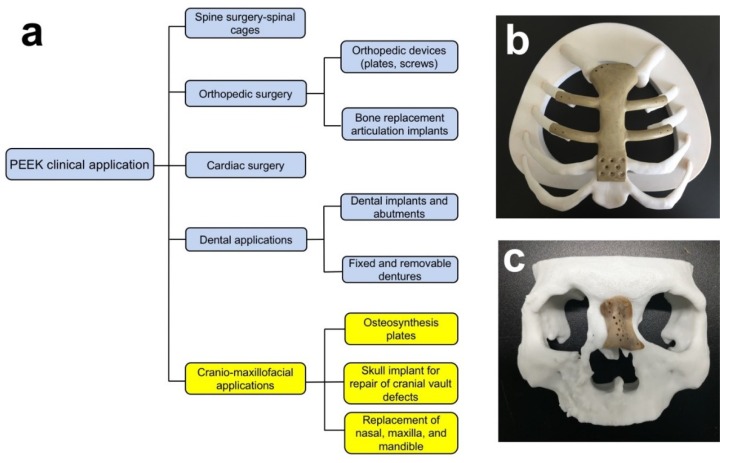
(**a**) Clinical applications of PEEK; Fused deposition modeling (FDM)-printed PEEK (**b**) breastbone and (**c**) nasal reconstructions.

**Figure 2 jcm-08-00240-f002:**
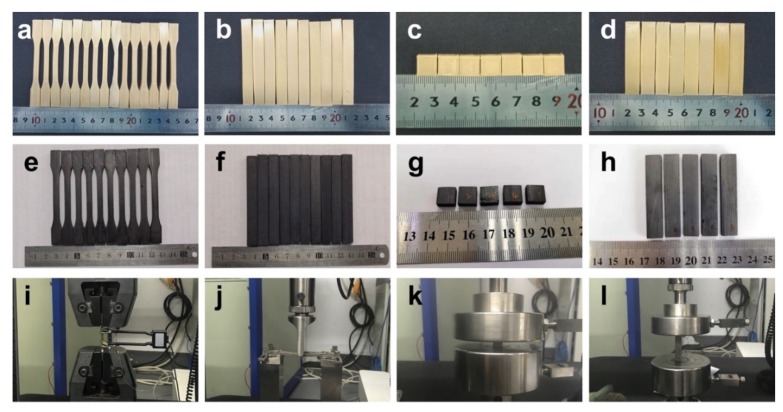
Pure PEEK and CFR-PEEK samples for testing mechanical properties: (**a**,**e**) tensile samples of PEEK and CFR-PEEK; (**b**,**f**) bending samples of PEEK and CFR-PEEK; (**c**,**g**) compressive samples (compressive strength) of PEEK and CFR-PEEK; (**d**,**h**) compressive samples (compressive modulus) of PEEK and CFR-PEEK; (**i**) tensile test; (**j**) bending test; (**k**,**l**) compressive tests of strength and modulus.

**Figure 3 jcm-08-00240-f003:**
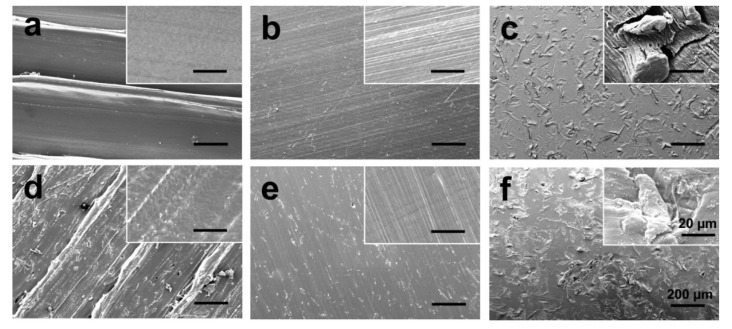
SEM images of PEEK and CFR-PEEK composite: (**a**) untreated PEEK; (**b**) polished PEEK; (**c**) sandblasted PEEK; (**d**) untreated CFR-PEEK; (**e**) polished CFR-PEEK; (**f**) sandblasted CFR-PEEK. Bars represent 200 µm and 20 µm (inserts), respectively.

**Figure 4 jcm-08-00240-f004:**
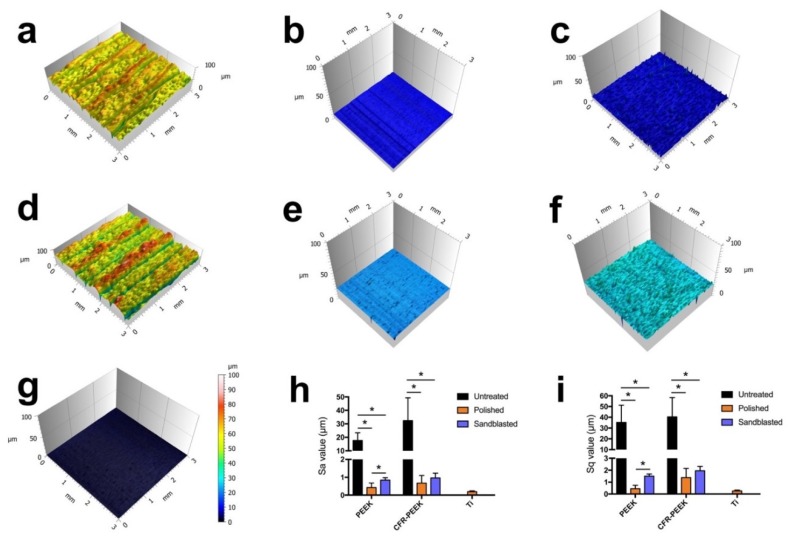
Reconstructed three-dimensional (3D) surface topographies of analyzed samples, and Sa and Sq values: (**a**) untreated PEEK; (**b**) polished PEEK; (**c**) sandblasted PEEK; (**d**) untreated CFR-PEEK; (**e**) polished CFR-PEEK; (**f**) sandblasted CFR-PEEK; (**g**) polished Ti; (**h**,**i**) Sa and Sq values of as-printed, polished, and sandblasted PEEK and CFR-PEEK samples, the polished Ti was used as an additional reference. The data are presented as means ± standard deviation, * *p* < 0.05.

**Figure 5 jcm-08-00240-f005:**
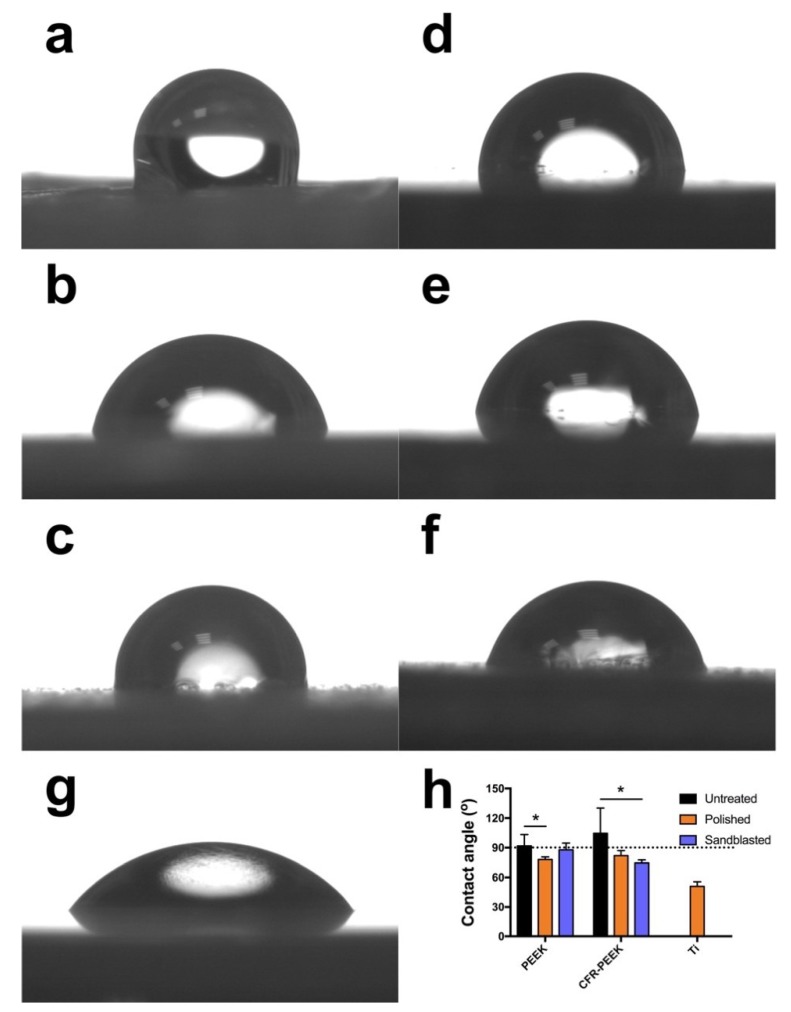
Water contact angle measured on untreated, polished, and sandblasted PEEK and CFR-PEEK samples: (**a**) untreated PEEK; (**b**) polished PEEK; (**c**) sandblasted PEEK; (**d**) untreated CFR-PEEK; (**e**) polished CFR-PEEK; (**f**) sandblasted CFR-PEEK. (**g**) Ti (additional reference); (**h**) quantitative contact angle values (means ± standard deviation). The dotted line indicates the contact angle of 90°, which is the division of hydrophilicity and hydrophobicity, * *p* < 0.05.

**Figure 6 jcm-08-00240-f006:**
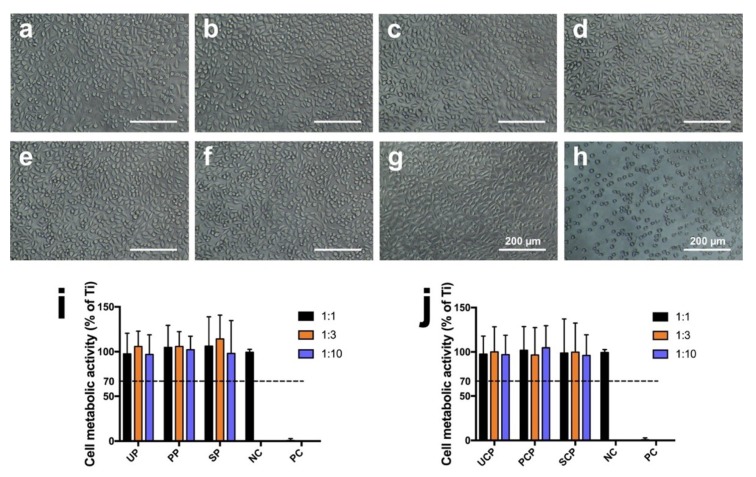
Cytotoxicity tests of L929 cells of PEEK and CFR-PEEK 100% extracts: (**a**) untreated PEEK; (**b**) polished PEEK; (**c**) sandblasted PEEK; (**d**) untreated CFR-PEEK; (**e**) polished CFR-PEEK; (**f**) sandblasted CFR-PEEK; (**g**) negative control (Ti); (**h**) positive control (Cu). (**i**,**j**) shows the quantitative result of the CCK-8 test in the culture media with different extract concentrations of PEEK and CFR-PEEK. The data are presented as means ± standard deviation. UP: untreated PEEK; PP: polished PEEK; SP: sandblasted PEEK; NC: negative control; PC: positive control; UCP: untreated CFR-PEEK; PCP: polished CFR-PEEK; SCP: sandblasted CFR-PEEK. The dotted line indicates the toxicity threshold of 70% cell viability according to ISO 10993-5.

**Figure 7 jcm-08-00240-f007:**
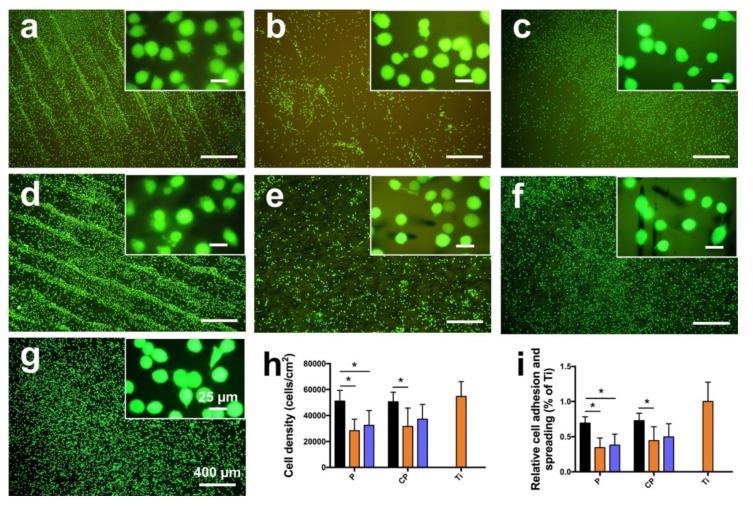
LIVE/DEAD staining of L929 cells on PEEK and CFR-PEEK samples after culturing for 24 h, with Ti as an additional control. (**a**) untreated PEEK; (**b**) polished PEEK; (**c**) sandblasted PEEK; (**d**) untreated CFR-PEEK; (**e**) polished CFR-PEEK; (**f**) sandblasted CFR-PEEK; (**g**) Ti. (**h**,**i**) shows the quantitative cell density and quantification of the mean surface area covered by cells. The data are presented as means ± standard deviation, * *p* < 0.05. P: PEEK; CP: CFR-PEEK; black bar: untreated group; orange bar: polished group; blue bar: sandblasted group. Cytotoxic effects, indicated by dead (red stained) cells, are not detectable.

**Figure 8 jcm-08-00240-f008:**
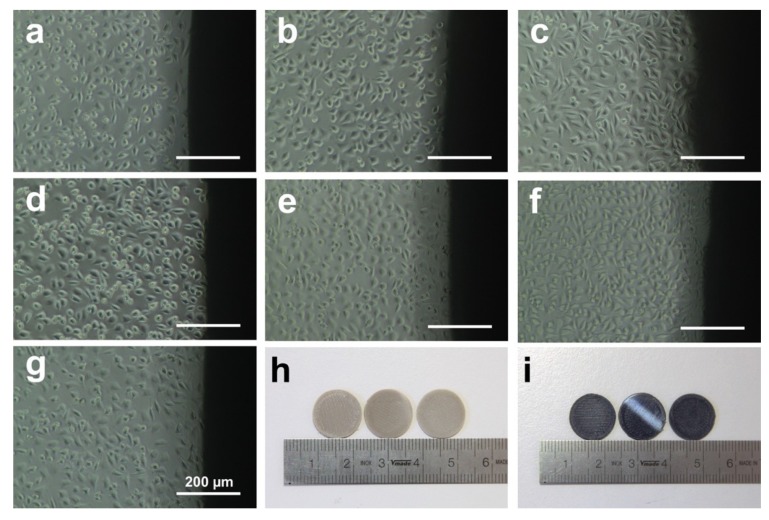
Microscopic images of L929 cells observed around samples of direct contact test after culturing for 24 h. (**a**) untreated PEEK; (**b**) polished PEEK; (**c**) sandblasted PEEK; (**d**) untreated CFR-PEEK; (**e**) polished CFR-PEEK; (**f**) sandblasted CFR-PEEK; (**g**) Ti; (**h**) PEEK samples (untreated, polished, and sandblasted); (**i**) CFR-PEEK samples (untreated, polished, and sandblasted).

**Figure 9 jcm-08-00240-f009:**
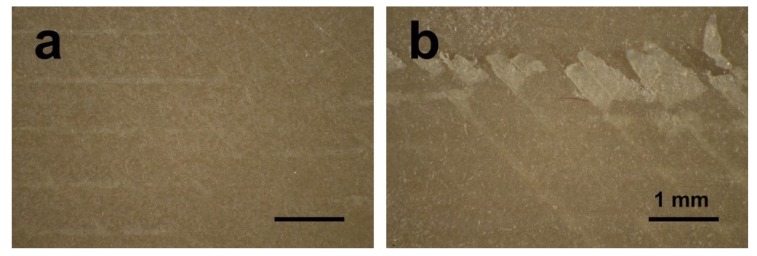
Optical micrographs of sandblasted PEEK samples (**a**) under 0.1 MPa pressure; (**b**) under 0.5 MPa pressure.

**Table 1 jcm-08-00240-t001:** The elastic modulus of different materials and human tissues.

Materials	Elastic Modulus (GPa)	References
PEEK	3–4	[8]
Ti	102–110	[8]
Zirconia	210	[15]
Cortical bone	14	[8]
Trabecular bone	1	[15]

**Table 2 jcm-08-00240-t002:** Technical specifications of the FDM printer.

Parameters	Technical Specifications
Nozzle diameter	0.4 mm
Bead width	0.4 mm
Layer thickness	0.2 mm
Printing speed	40 mm/s
Raster angle	Consistent with the longest edge
Ambient temperature	20 °C
Nozzle temperature	420 °C

**Table 3 jcm-08-00240-t003:** Mechanical properties (means ± standard deviation) of PEEK and CFR-PEEK.

Groups	Tensile Strength (MPa)	Tensile Modulus (GPa)	Bending Strength (MPa)	Bending Modulus (GPa)	Compressive Strength (MPa)	Compressive Modulus (GPa)
**PEEK**	95.21 ± 1.86 ^a^	3.79 ± 0.27 ^a^	140.83 ± 1.97 ^a^	3.56 ± 0.13 ^a^	138.63 ± 2.69 ^a^	2.79 ± 0.11 ^a^
**CFR-PEEK**	101.41 ± 4.23 ^b^	7.37 ± 1.22 ^b^	159.25 ± 13.54 ^b^	5.41 ± 0.51 ^b^	137.11 ± 3.43 ^a^	3.51 ± 2.12 ^b^

Different lowercase letters in the same column indicate significantly different groups (*p* < 0.05).

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
