# Peer review of "Carbon Fiber Reinforced PEEK Composites Based on 3D-Printing Technology for Orthopedic and Dental Applications"

_jcm, 2019, doi:10.3390/jcm8020240_

Round 1
Reviewer 1 Report
Dear Authors,
your paper is the result of an huge amount of work and the methods that you used are clearly described with a number of interesting details. Nevertheless, I have to remark that the abundance of details in the Introduction and Results make the reading of your work cumbersome, and decrease the quality of the excellent work performed in the lab. CFR-PEEK has been investigated since a long (see e.g. Scotchford CA, et al. Biomaterials 2003:4871-9, Sagomonyants KB,et al. Biomaterials 2008:1563-72) and its biological safety in load bearing applications (e.g. bone plates for osteosynthesis) is acknowledged.
Your article reports nice experimental work made in production and testing CFR-PEEK by AM by FDM. While the wettability test is indicative of the potential advantage of the as-made CFR-PEEK surface among the ones tested, it is is unclear what the cytotoxicity tests are add to the overall conclusion of your paper. It is noted that your paper underlines in several points the relevance of the demonstration of the non-cytotoxicity of CFR-PEEK obtained by fused deposition modeling (FDM). Nevertheless, is is unclear why and in which way FDM may change the biological safety of CFR-PEEK in comparison with other technologies used for obtaining CFR-PEEK devices. Would you please clarify this point.
As you remark in the paper Introduction PEEK is not bioresorbable nor biodegradable, and so is CFR-PEEK. This behavior has limited albeit excluded its use in tissue engineering and in bone regenerative medicine. In addition, today synthetic bone grafts are selected on their osteoconductivity, while CFR-PEEK link to the host bone is due to mechanical interlock only, unless treated to make its surface bioactive.
The revision of this paper is recommended, addressing the working hypothesis: is the material suitable for the development of which specific device (dental implant? Bone plate?....). A more concise style of presentation is recommended.
In addition, please note:
Line 228/90 Ck Table number & Content Vs. text.
Line 344 Ck Table number.
Line 348: ….still insufficient. Please clearify for what? For which application?
Line 441/4 To be incorporate in the Conclusions.
Author Response
Thank you very much for reviewing our manuscript. Your suggestions are helpful for us to improve the quality of the paper. We have already revised our manuscript according to your comments, including some details you pointed out.

Reviewer 2 Report
Comments to the Authors:
The objective of this study was to evaluate the mechanical properties of FDM type 3D printing PEEK and CFR-PEEK. And the influence for FDM-printed PEEK and CFR-PEEK with surface treated by polishing or sandblasting were characterized for surface roughness, biocompatibility, and cell adhesion. Some suggestions are listed as following:
1. In Title, (a) Carbon Fiber Reinforced PEEK Composite → Carbon Fiber Reinforced PEEK Composites (b) for Orthopedic and Dental Application → for Orthopedic and Dental Applications
2. In abstract, it was claimed that the major issues for FDM PEEK objects were difficult to process due to high melting and limit to clinical application due to bioinertness. However, the experimental design were directed to mechanical tests and biocompatibility for FDM-printed PEEK and CFR-PEEK. Please offer more clear rationales about the reason why doing this study.
3. In abstract, it was claimed that the results indicated that the printed CFR-PEEK samples had significantly higher mechanical strengths than the printed pure PEEK. However, the compressive strength for FDM-printed PEEK and CFR-PEEK were 138.63 ± 2.69 and 137.11 ± 3.43 MPa, respectively. The results is against the conclusions.
4. In Figure 2, please offer the pictures for CFR-PEEK objects.
5. According to ISO 604, the typical specimens for compressive strength measurement can either be blocks (12.7 x 12.7 x 25.4mm) or cylinders (12.7mm in diameter and 25.4mm in length. The samples shown in Figure 2-c for compressive strength were not the right size form ISO 604 guidance.
6. According to ISO 25178, the quantitative analysis of surface texture includes the measurement of roughness, form and waviness. In the study, the AFM was used to measure Ra, known as the term of roughness, but the data here also included the contribution form and waviness in Figure 4-(a) and 4-(d). In addition, the surface roughness is nothing to do with cytotoxicity test since the specimens were prepared by polar or non-polar solvent extraction.
7. Updated references such as following study should be included. А.А. Stepashkin, D.I. Chukov, F.S. Senatov, A.I. Salimon, A.M. Korsunsky, S.D. Kaloshkin, 3D-printed PEEK-carbon fiber (CF) composites: Structure and thermal properties, Composites Science and Technology, Vol 164, 2018, Pages 319-326.
8. Please offer your discussion on why the less contact angle results were obtained for sandblasted PEEK and CFR-PEEK than those from untreated specimens?
Author Response
Thank you very much for reviewing our manuscript. Your suggestions are helpful for us to improve our paper. We have already revised it according to your comments, including introduction, methods, results, discussion, and conclusion parts. Besides, our paper has been thoroughly edited by a native English speaking colleague.

Reviewer 3 Report
Dear Authors
The paper needs some changes.
Why you wrote that PEEK is a polyaromatic semi-crystalline thermoplastic polymer with an elastic modulus of 3-4 GPa (Table 1), which is much lower than that of Ti (102-110 GPa) and very close to the human trabecular bone (1 GPa). ? Can you explain this difference with lower paper?
The Elastic modulus of cortical bone was reported to be approximately 10–30 GPa, and that of Ti alloysv50 such as Ti-6Al-4V was approximately 110 GPa [8]. Long, M.; Rack, H.J. Titanium alloys in total joint replacement—a materials science perspective. Biomaterials 1998, 19, 1621–1639.
Dear auhor please include a photo of this machine electro-hydraulic servo mechanical testing machine (CMT4304, MTS Corp., USA)
Can you explain why you didn’t used the osteoblasts? This is a bone forming cell Is more important than fibroblasts.. Thanks
Other question. Did you desinfected the tested materials before cell culture?
The discussion section is well written.
Author Response
Thank you very much for reviewing our manuscript. Your comments are helpful for us to improve our paper. We have revised our manuscript according to your suggestions.

Reviewer 4 Report
Interesting topic and a highly useful manufacturing technique of PEEK components.
Review English language and style.
I have listed some more specific additional comments below. Row 49: Used widely feels a bit overstated. Investigated widely seems more likely. Row 123: How were the Ti disc manufactured? Subtractive or additive manufacturing? Row 155: Were the discs sputtered prior SEM analysis? Row 343: Since PEEK in some applications is a more esthetic alternative to Ti. How was the coloration changed with 5% of carbon.
Author Response
Thank you very much for reviewing our manuscript. Your comments are very helpful for us to improve the quality of our paper. We have already revised our paper according to your suggestions, including English editing and other necessary changes.

Round 2
Reviewer 1 Report
Dear Authors,
Your paper has been significantly improved, and reports a significant amount of activity in process characterization. The amount of work performed in biological characterization is surprising, taken into account that it is due to the use of a fixative paper: why not characterize the paper before accepting it in your process? This question will not have an answer, but I suggest a better planning of your future development work.
Reviewer 2 Report
It looks OK to me.